# Comparison of Diagnosed Depression and Self-Reported Depression Symptom as a Risk Factor of Periodontitis: Analysis of 2016–2018 Korean National Health and Nutrition Examination Survey Data

**DOI:** 10.3390/ijerph18030871

**Published:** 2021-01-20

**Authors:** Seon-Rye Kim, Seoul-Hee Nam

**Affiliations:** 1Department of Pharmacy, College of Pharmacy, Kangwon National University, 1 Gangwondaehakgil, Chuncheon-si, Gangwon-do 24341, Korea; sjsanj@hanmail.net; 2Department of Dental Hygiene, College of Health Science, Kangwon National University, 346 Hwangjo-gil, Dogye-up, Samcheok-si, Gangwon-do 25945, Korea

**Keywords:** depression, periodontitis, national health insurance, sleeping hours

## Abstract

Depression causes damage to the immune defense mechanism, and it can worsen periodontitis due to the accumulation of periodontitis pathogens. This study was conducted in order to explore the association of diagnosed depression and self-reported depression symptom with periodontitis by using the Korean National Health and Nutrition Examination Survey 7th (KNHANES VII) data. A total of 12,689 participants aged over 19 received a periodontal examination among the 24,269 participants of KNHANES VII. Diagnosed depression and self-reported depression symptom were the two terms used for depression. Periodontitis was defined as the presence of teeth with periodontal pockets of 4 mm or deeper. The age, sex, marital status, education, region, basic livelihood protection, private health insurance, type of housing, health insurance coverage, household income, sleeping hours, subjective health condition, stress perception, drinking status, obesity, and current smoking status of the participants were examined. Chi-square tests and two-tailed analyses were used. The association of depression and periodontitis was tested by using logistic regression models adjusted for socio-demographic and behavioral variables. Diagnosed depression was associated with periodontitis, as the odds ratio of diagnosed depression for periodontitis was 1.772 (95% confidence interval = 1.328–2.364). However, the association between self-reported depression symptom and periodontitis was not statistically significant. This study revealed that diagnosed depression, not self-reported depression symptom, could be a risk factor for periodontitis. Therefore, it is necessary to take a closer look into diagnosed depression in order to manage and prevent periodontitis.

## 1. Introduction

The oral cavity not only protects the human body from systematic infection, and enables chewing and swallowing in a biological perspective, but also plays an important role in self-evaluation, expression, communication, and esthetics in a psychosocial perspective [1]. Oral disease is a global primary public health problem that affects individuals and society through pain, oral function disorder, and decrease in quality of life [2]. Periodontitis is a common disease of the dental supporting apparatus in which the periodontal tissue is inflamed and deteriorated, and causes pathological alveolar bone loss, loose tooth, and abscess, which is a major cause of tooth loss in Korean adults [3]. It has been reported that the occurrence and progress of periodontitis are influenced by a combination of various factors, which primarily consist of bacteria, followed by gender, age, psychological factors, genetic factors, smoking, systemic diseases, socio-economic factors, and education [4].

Resistance to disease or external stress deteriorates as the function and production of immune substances declines with aging [3]. Periodontitis is reported to affect the immune system through mental conditions, such as stress and anxiety, thereby making a person vulnerable to infection and other diseases [5]. For this reason, periodontitis is considered a complex disease [6]. Depression is also associated with the inhibition of mitogen-induced lymphocyte hyperplasia. Furthermore, it affects the number and function of immune cells [7], and the deterioration of immune defense caused by this results in the accumulation of pathogens, which leads to a more severe periodontitis [8]. 

The prevalence rate of depression is 7.8% to 17.1% globally [9] and 5.6% for Korea [10], and it is common enough to compose 10% of primary hospital visits [11]. Depression damages a person’s social and physical wellbeing more than any chronic diseases, including hypertension, diabetes, arthritis, and backache [12]. In particular, Korean depression patients tend to complain more about their physical condition than their sense of depression [13]. Araújo et al. showed no significant association between depression and periodontitis [14]. Nascimento et al. conducted a clinical study on 539 subjects and reported that there was a statistically significant relation between susceptibility to depression symptoms and periodontitis. Neither the presence of depressive symptoms nor the presence of major depression was associated with the combination of clinical attachment loss (CAL) and bleeding on probing (BOP) [15]. On the other hand, a study has shown that oral health has a significant effect on depression [16]. 

Since inconsistencies are observed in studies on the association between depression factors and periodontitis, research on psychological factors as risk factors for persistent periodontitis is deemed necessary. However, few studies have been conducted in order to determine whether the definition of depression changes the association between depression and periodontitis. So, we formulated a hypothesis that diagnosed depression and self-reported depression symptom would be related to periodontitis

Therefore, this study used the Korean National Health and Nutrition Examination Survey data that represent the situation in Korea to determine whether depression classified into self-reported depression and diagnosed depression shows association with periodontitis in order to contribute to oral health management and the development of medical policy solutions. 

## 2. Materials and Methods

### 2.1. Study Subjects

The Korean National Health and Nutrition Examination Survey (KNHANES), which is performed annually by the Korea Center for Disease Control and Prevention (KCDC) to measure health status and nutrition status of the South Korean population, is a series survey of national representative samples. This study used data from KNHANES VII conducted during 2016–2018. The sampling method for the KNHANES VII was designed to include a complex, stratified, multistage, probability cluster survey of a representative sample of a civilian population. KNHANES VII was conducted by obtaining the approval of Research Ethics Review Committee of KCDC for the 3rd year (2018-01-03-P-A). The survey employed stratified multistage probability sampling units based on geographic area, gender, and age, which were determined based on the household registries of the 2010 National Census Registry, the most recent 5-year national census in Korea. Using the 2010 census data, 576 primary sampling units (PSU) were selected across Korea. The inclusion criteria in the study are (i) who are final sample individuals for KNHANES VII, (ii) who participate KNHANES VII. The exclusion criteria in this study are (i) who are under 19 years old, (ii) who did not receive a periodontal examination. Figure 1 shows the flowchart for inclusion and exclusion.

### 2.2. Socio-Demographic and Health-Behavioral Characteristics

The socio-demographic variables included sex, age, marital status, education, region, type of housing, household income, experience of basic livelihood protection, health insurance coverage, and private health insurance. Age was classified into seven categories: <30, 30–39, 40–49, 50–59, 60–69, and >=70. Marital status was classified into living with a spouse and living alone. Region was divided into Seoul, metropolitan city, Gyeonggi province, and remaining 7 provinces. Education was classified into above high school degree or below high school degree. Types of housing were divided into detached house and apartments. Household income was divided into more than $30K or less than $30K. Health insurance coverage was classified into local health insurance, employment-based insurance, and Medicaid. 

General health behavior variables included current smoking status, drinking status, sleeping hours, obesity, subjective health condition, and stress perception. Sleeping hours was divided into under 8 h a day or above 8 h a day. Obesity was classified into three groups: underweight (<BMI 18.5 kg/m^2^,), normal (BMI 18.5 kg/m^2^–BMI 24.99 kg/m^2^), and obesity (BMI 25 kg/m^2^ =<). Subjective health condition was categorized into three groups: good, normal, and bad. For drinking status, score 1 was given for drinking once or more per month for the recent 1 year, and 0 if not. 

### 2.3. Periodontitis

The presence of periodontitis was measured from the oral examination data, such as community periodontal index (CPI). For the CPI, the mouth was divided into sextants by tooth numbers: 18–14, 13–23, 24–28, 38–34, 33–43, and 44–48. A sextant was examined only if there were two or more teeth present that were not indicated for extraction (WHO, 1997). Periodontal status was measured by using the ball probe designed by WHO. Code 0 means healthy; Code 1 means bleeding after probing; Code 2 means calculus; Code 3 means 4 to 5 mm periodontal pocket; and Code 4 means above 6 mm periodontal pocket. In this study, a score over 3 was defined as periodontitis based on KNHANES VII. The periodontitis measurement was performed by dentists [17]. The dentists calibrated periodontal status. Oral examinations were conducted by trained dentists in compliance with the World Health Organization oral examination criteria. In field training for quality control of oral examinations, Kappa values were calculated to verify the consistency of oral test results for the reliability of the investigators. 

### 2.4. Types of Depression

Two kinds of depression were used in this study. The first depression was diagnosed depression. It was assessed by a trained interviewer asking the participants: “Have you ever been diagnosed with depression by a physician?” The responses were “No” or “Yes”.

The second depression was self-reported depression symptom. It was measured using the Patient Health Questionnaire (PHQ-9), which assesses the frequency of diagnostic criteria of depression during the past two weeks. The scores of nine diagnostic criteria of PHQ-9 were added. If the sum score is above 10, the individuals were considered as people with self-reported depression symptom, producing a sensitivity of 81% and a specificity of 82% for any depression [18]. 

### 2.5. Statistical Analyses

We applied the stratification variable, clustering variable, and weight for all analyses. Complex samples chi-square tests were carried out in order to compare diagnosed depression and self-reported depression symptom according to socio-demographic and behavioral variables. In order to analyze the effects of the socio-demographic variables, behavioral variables, and depression on presence of periodontitis, complex samples logistic regression analysis was carried out. Model I was analyzed by integrating the socio-demographic variables, and behavioral variables. Model II was analyzed by integrating diagnosed depression and self-reported depression symptom. Model III was analyzed by integrating socio-demographic variables, behavioral variables, and depression. SPSS ver. 21.0 was used to analyze data, and the statistical significance was *p* < 0.05, two-tailed. 

## 3. Results

### 3.1. Socio-Demographic Variables and Behavioral Variables by Periodontitis 

The final sample set for KNHANES included 13,248 households. Among 31,689 sampled individuals, the number of participants was 24,269. The response rate was 76.6%. A total of 19,389 individuals aged over 19 participated in KNHANES, but 12,689 of the participants received a periodontal examination. A detailed description of the sampling was described in the KNHANES report. 

Concerning socio-demographic variables, the proportion of male who have periodontitis was 37.7% and higher than that of female (*p* = 0.000). About age, the proportion of people above 70 years old with periodontitis was the highest as 47.9% (*p* = 0.000). People with periodontitis tend to live alone without a spouse (45.5%, *p* = 0.000), be low educated (47.8%, *p* = 0.000), live in other provinces except Gyeonggi-do, have Medical Aid program coverage (40.7%, *p* = 0.000), live in detached home, earn less than 30,000 dollars per year (40.1%, *p* = 0.000), have experience to get basic livelihood protection (39.6%, *p* = 0.000), and have no private health insurance (42.4%, *p* = 0.000).

Concerning behavioral variables, people with periodontitis tend to be obese, get stressed a lot, currently smoke, be unhealthy by subjective perception, and have diagnosed depression. There were significant differences only in sex, age, marital status, education, region, experience of basic livelihood protection, private health insurance, type of housing, health insurance coverage, household income, obesity, subjective health condition, stress perception, current smoking status, and diagnosed depression. (Table 1).

### 3.2. Socio-Demographic Variables and Behavioral Variables by Diagnosed Depression

Concerning sex, the proportion of female in diagnosed depression was 6.5% and higher than those of male (*p* = 0.000). About age, the proportion of people above 70 years old in diagnosed depression was the highest at 7.2% (*p* = 0.000). People with diagnosed depression tend to live alone without a spouse (8.9%, *p* = 0.000), be low educated (7.4%, *p* = 0.000), have Medical Aid program coverage (17.2%, *p* = 0.000), earn less than 30,000 dollars per year (6.9%, *p* = 0.000), have experience to get basic livelihood protection (12.8%, *p* = 0.000), and have no private health insurance (7.5%, *p* = 0.000).

Concerning behavioral variables, people with diagnosed depression tend to be underweight, sleep less than 8 h per day, get stressed a lot, currently not smoke, drink during current one month, and be unhealthy by subjective perception. There were significant differences only in sex, age, marital status, education, basic livelihood protection, private health insurance, health insurance coverage, household income, subjective health condition, stress perception, and drinking status (*p* < 0.001) (Table 2).

### 3.3. Socio-Demographic Variables and Behavioral Variables by Self-Reported Depression Symptom

Concerning socio-demographic variables, people with self-reported depression symptom tend to be female, be under 30 years old or above 60 years old, live alone without a spouse, be low educated, live in other provinces except Gyeonggi-do, live in a detached home, have Medical Aid program coverage, earn under 30,000 dollars per year, have experience to get basic livelihood protection, and have no private health insurance.

Concerning behavioral variables, people with self-reported depression symptom tend to be underweight, sleep under 8 h per day, get stressed a lot, currently smoke, drink during current one month, and be unhealthy by subjective perception. There were significant differences only in sex, age, marital status, education, region, basic livelihood protection, private health insurance, type of housing, health insurance coverage, household income, obesity, subjective health condition, stress perception, drinking status, and current smoking status (Table 3).

### 3.4. Diagnosed Depression and Self-Reported Depression Symptom

The proportion of persons who have self-reported depression symptom with diagnosed depression was 86.9% and higher than that of persons who do not have self-reported depression symptom with diagnosed depression (*p* = 0.000) (Table 4).

### 3.5. Effects of Socio-Demographic Variables, Behavioral Variables, and Depression on the Presence of Periodontitis

Model I was analyzed by integrating socio-demographic variables and behavioral variables. In terms of epidemiological characteristics, male (OR: 1.690, 95% CI: 1.463–1.954), the 60–69 aged (OR: 9.196, 95% CI: 4.257–19.866), living with a spouse (OR: 0.734, 95% CI: 0.606–0.889), below middle school graduate (OR: 1.269, 95% CI: 1.065–1.514) living in Gyeonggi-do province (OR: 0.600, 95% CI: 0.468–0.768), and having the employee insured (OR: 1.260, 95% CI: 0.791–2.007) were associated with risk of presence of periodontitis. For health-related variables, obesity (OR: 1.262, 95% CI: 1.104–1.443) and current smoking status (OR: 2.143, 95% CI: 1.783–2.575) increased risk of presence of periodontitis.

Model II was a result of unadjusted logistic regression of diagnosed depression and self-reported depression symptom for periodontitis. Diagnosed depression (OR: 1.443, 95% CI: 1.079–1.898) increased risk of presence of periodontitis, and self-reported depression symptom (OR: 1.133, 95% CI: 0.857–1.497) increased risk of presence of periodontitis

Model III was analyzed by integrating socio-demographic variables, behavioral variables, diagnosed depression, and self-reported depression symptom. After adjusting socio-demographic variables and behavioral variables, diagnosed depression (OR: 1.772, 95% CI: 1.328–2.364) and self-reported depression symptom (OR: 1.250, 95% CI: 0.875–1.787) increased risk of presence of periodontitis (Table 5).

## 4. Discussion

Recently, the difference between average life expectancy and health life expectancy has led to the need for a quality of life in living long and in good health [19]. Considering the importance of oral health and since the mouth is physically, mentally, and socially affected just like any other organ, general health cannot be maintained without including oral health [20]. In addition, the association between poor oral health and systematic disease is a very important issue that needs to be addressed [21].

In a representative sample of the Korea population, we found that people diagnosed with depression in the past year were almost twice as likely to have periodontitis than those without diagnosed depression. This relationship was valid after the adjustment of socio-demographic variables and behavioral variables. Our analyses also revealed that living region, type of housing, health insurance coverage, and current smoking status were related to periodontitis. In contrast to diagnosed depression, self-reported depression symptom was not associated with periodontitis. Our findings suggest the possibility that diagnosed depression, but perhaps not self-reported depression symptom, may enhance the risk of periodontitis.

The purpose of this study is to verify the association between depression and periodontitis. One of these results, which showed that self-reported depression symptom was not associated with periodontitis, was consistent with those of Castro et al. [22], who found that self-reported depression symptom severity was not associated with periodontal indicators in a sample of 165 adults. It was reported that depression and general anxiety are not associated with periodontitis [23], and for geriatric patients, it was reported that depression is associated with the change in facial appearance due to tooth loss and pain-related chronic disease, but not with periodontitis [24].

The other result, which showed that diagnosed depression was associated with periodontitis, was consistent with various research results. Saletu A. et al. reported that the periodontitis rate increased with the increase in psychological factors, including depression and anxiety, and decrease in the quality of life [25]. Ng SK et al. reported that these factors were also associated with clinical attachment loss [26]. In addition, the factors contributed to the progress of acute progressive periodontitis in adults [27]. Stress has been reported to increase with the decrease in oral health and results in an increase in depression symptoms [26,28,29]. The result showing that diagnosed depression is associated with the outbreak of periodontitis, while current depression symptom is not associated with periodontitis, demonstrates that periodontitis is a result of chronic infection. It also shows that the past depression diagnosis has a greater impact on the development of chronic periodontitis than the current depression symptoms. Therefore, if diagnosed depression, not depression symptoms, is used as a variable in the study on the association between depression and periodontitis, a clear association with periodontitis will be observed.

Clinical research shows the causal relationships between periodontitis and depression [15,27]. However, it is difficult to find a result that clearly demonstrates association between depression and periodontitis among the studies that are based on the national survey data. In addition, many studies have attempted to verify the association between depression and periodontitis after stratifying patients according to their age [16,23,30,31,32]. In this study, an association was found between depression and periodontitis in all age groups in cases where depression was diagnosed. However, depression symptoms showed no association with periodontitis. This result differed from Khambaty et al. [23] and Delgado-Angulo et al. [31], who reported that there was no connection between periodontitis and depression. In the univariate analyses of risk factors on periodontitis, it was found that socio-demographic variables (e.g., sex, age, marital status, education, house income, region, type of housing, health insurance coverage, and private health insurance) were associated with the presence of periodontitis in all age group. In addition, the association has also been seen in a multiple regression analysis for some socio-demographic variables, including marital status, education, region, type of housing, and health insurance coverage.

For marital status, the occurrence rate of periodontitis was low in living with a spouse cases, which may indicate that the care of the spouse living together is relevant [33]. Additionally, different from Thomson et al., who reported that low income was associated with periodontitis and depression [34], there was no significant difference found in different income levels. This may be influenced by the criteria for classifying income levels. However, living in apartments showed a low periodontitis occurrence rate than living in a detached housing. This can be interpreted as a low socioeconomic status associated with the severity of periodontitis and depression, as individuals living in apartments tend to have a higher economic position in Korea. The lower periodontitis occurrence for employment-based insurance covered cases compared to Medicaid covered cases could be interpreted in the same context as above. In addition, Seoul and Gyeonggi province showed lower periodontitis rate than the rest of the regions. It is considered that this difference is linked to the difference in financial independence, i.e., the socioeconomic level of the region. The association between low education level and the high rate of periodontitis shares the same context with Eke et al., who reported that low education level is associated with the development of periodontitis and the severity of depression [35].

Health-related behavioral variables may explain the association between diagnosed depression and periodontitis. For instance, we found that obesity and current smoking status were related to periodontitis. These findings suggest that obesity and current smoking status can increase the prevalence of periodontitis [36]. Other health-related behavioral variables (e.g., sleeping hours, stress, subjective health status, and current drinking status) were not associated with diagnosed depression. These results are not consistent with those of Genco and Borgnakke [37]. Moreover, these results are not consistent with those of Grover et al., which showed the causal relationships between sleep deprivation and periodontitis [38]. In addition, our study did not reveal that stress perception was connected with periodontitis [37].

Psychological factors have been reported to contribute to the pathogenesis of inflammatory periodontitis [39], and it was suggested that a link was found between psychosocial factors and periodontitis [40]. Stress and depression can contribute to the outbreak and progress of periodontitis through change in health-related behaviors and physiological mechanisms. Depression reduces the frequency of oral health actions, thereby resulting in loss of attachment and loss of teeth. Chronic stress and depression deteriorate the cellular and humoral immune system and weaken the resistance against periodontal tissue destruction [41]. Periodontitis is a microbially driven host-mediated slowly progressive destructive disease of the periodontium. Therefore, it was confirmed that the diagnosed depression was associated with chronic periodontitis that develops slowly due to prolonged depression.

Since studies on the link between depression and periodontitis have not reached a clear conclusion, this study attempted to identify the roles of depression as a risk factor on periodontitis. We divided the definition of stress into diagnosed depression and self-reported depression in order to identify their association with periodontitis. As a result, cases where depression was diagnosed had a higher prevalence rate of periodontitis compared to cases that were not diagnosed even when the various socio-demographic and health behavioral variables were adjusted. However, for self-reported depression, no association was found with periodontitis. This demonstrates that diagnosed depression, not the current depression symptoms, is a risk factor for periodontitis. It is deemed necessary to take a closer look into diagnosed depression in order to manage and prevent periodontitis.

This study has some limitations. As this is a cross-sectional study, causal relationships cannot be identified. As a follow-up study, it is necessary to discuss the occurrence of periodontal disease through accurately diagnosed depression based on this study. Further studies should be performed to confirm and generalize these results. In addition, we used only CPI results as periodontal disease, and excluded important data such as clinical attachment loss. Although diagnosed depression was assessed by asking questions, and self-reported depression symptom was measured by asking structured questionnaires, there could be the undervaluation of presence for depression.

## 5. Conclusions

Since diagnosed depression turned out to be a risk factor for periodontitis, it is suggested that periodontitis could be managed more efficiently in Korea through the development of policies to connect oral health improvement programs with mental health improvement programs.

## Figures and Tables

**Figure 1 ijerph-18-00871-f001:**
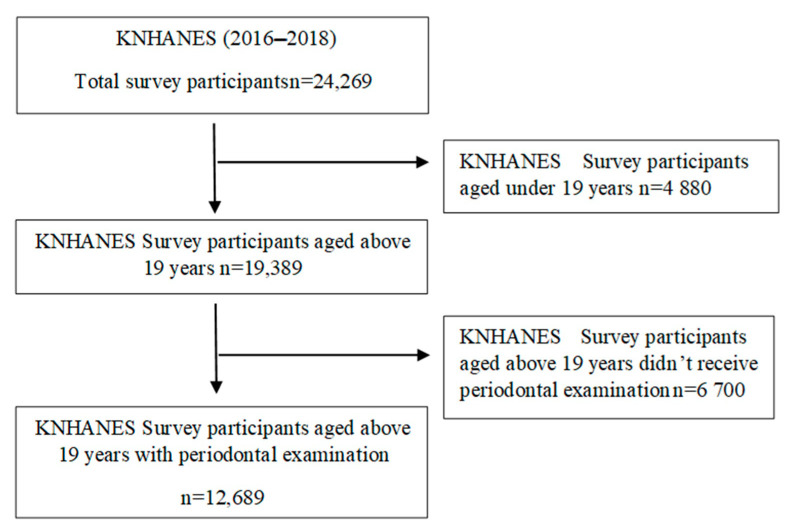
Flow chart.

**Table 1 ijerph-18-00871-t001:** Socio-demographic variables and behavioral variables by periodontitis.

Characteristics	Sub-Items	Periodontitis		
		No	Yes	*p*
Sex	Male	3417 (62.3)	2128 (37.7)	0.000 **
	Female	5291 (74.7)	1853 (25.3)	
Age	<30	1501 (96.0)	61 (4.0)	0.000 **
	30–39	1806 (86.5)	294 (13.5)	
	40–49	1768 (74.8)	647 (25.2)	
	50–59	1466 (60.6)	1005 (39.4)	
	60–69	1144 (53.7)	1017 (46.3)	
	70=<	1023 (52.1)	957 (47.9)	
Marital Status	Living with a spouse	5815 (67.1)	2975 (32.9)	0.000 **
	Living alone	932 (54.5)	782 (45.5)	
Education	=<Middle school	1826 (52.2)	1669 (47.8)	0.000 **
	High school=<	6513 (76.5)	2102 (23.5)	
Region	Seoul	1818 (74.1)	624 (25.9)	0.000 **
	Metropolitan	2567 (68.3)	1143 (31.7)	
	Gyeonggi-do province	1975 (76.4)	614 (23.6)	
	Other provinces except Gyeonggi-do	2348 (61.3)	1600 (38.7)	
Experience of Basic Livelihood Protection	Yes	517 (60.4)	372 (39.6)	0.000 **
	No	8186 (70.1)	3605 (29.9)	
Private Health Insurance	Yes	7099 (72.6)	2780 (27.4)	0.000 **
	No	1553 (57.6)	1175 (42.4)	
Type of Housing	Detached home	2617 (61.5)	1692 (38.5)	0.000 **
	Apartment house	6090 (73.5)	2289 (26.5)	
Health Insurance Coverage	Self-employed (insured)	2333 (64.1)	1328 (35.9)	0.000 **
	Employee (insured)	6051 (72.3)	2426 (27.7)	
	Medical Aid program	272 (59.3)	191 (40.7)	
Household Income	<30,000 dollars	2817 (59.9)	1912 (40.1)	0.000 **
	30,000 dollars=<	5870 (74.9)	2054 (25.1)	
Sleeping Hours	<8 h per day	5280 (69.3)	2437 (30.7)	0.269
	8 h per day=<	3051 (70.3)	1319 (29.7)	
Obesity	Underweight	376 (81.9)	95 (18.1)	0.000 **
	Normal	4181 (73.5)	1548 (26.5)	
	Obese	2124 (64.1)	1189 (35.9)	
Subjective Health Condition	Good	2547 (74.0)	895 (26.0)	0.000 **
	Normal	4373 (69.6)	1994 (30.4)	
	Bad	1463 (61.6)	913 (38.4)	
Stress Perception	No	7356 (71.2)	3111 (28.8)	0.000 **
	Yes	1277 (60.7)	824 (39.3)	
Drinking Status	No	3921 (68.8)	1830 (31.2)	0.152
	Yes	4716 (70.0)	2113 (30.0)	
Current Smoking Status	No	7344 (71.8)	2966 (28.2)	0.000 **
	Yes	1292 (58.2)	969 (41.8)	
Diagnosed Depression	No	7975 (69.3)	3636 (30.7)	0.032 *
	Yes	364 (65.1)	195 (34.9)	
Self-reported Depression Symptom	No	5292 (70.5)	2266 (29.5)	0.738
	Yes	287 (69.5)	138 (30.5)	

* *p* < 0.05, ** *p* < 0.001.

**Table 2 ijerph-18-00871-t002:** Socio-demographic variables and behavioral variables by diagnosed depression.

Characteristics	Sub-Items	Diagnosed Depression		
		No	Yes	*p*
Sex	Male	5391 (97.7)	133 (2.3)	0.000 **
	Female	6667 (93.5)	444 (6.5)	
Age	<30	1476 (96.6)	51 (3.4)	0.000 **
	30–39	1949 (96.9)	54 (3.1)	
	40–49	2225 (96.2)	84 (3.8)	
	50–59	2296 (95.7)	95 (4.3)	
	60–69	2023 (94.0)	143 (6.0)	
	70=<	2089 (92.8)	150 (7.2)	
Marital Status	Living with a spouse	8343 (95.9)	340 (4.1)	0.000 **
	Living alone	1665 (91.1)	159 (8.9)	
Education	=<Middle school	3583 (92.6)	279 (7.4)	0.000 **
	High school=<	8409 (96.5)	295 (3.5)	
Region	Seoul	2353 (95.7)	100 (4.3)	0.111
	Metropolitan	3504 (95.4)	169 (4.6)	
	Gyeonggi-do province	2483 (95.9)	107 (4.1)	
	Other provinces except Gyeonggi-do	3718 (94.4)	201 (5.6)	
Experience of Basic Livelihood Protection	Yes	803 (87.2)	110 (12.8)	0.000 **
	No	11250(95.9)	467 (4.1)	
Private Health Insurance	Yes	9284 (96.1)	353 (3.9)	0.000 **
	No	2703 (92.5)	220 (7.5)	
Type of Housing	Detached home	4186 (94.8)	223 (5.2)	0.113
	Apartment house	7872 (95.6)	354 (4.4)	
Health Insurance Coverage	Self-employed (insured)	3538 (95.7)	153 (4.3)	0.000 **
	Employee (insured)	8045 (95.8)	335 (4.2)	
	Medical Aid program	399 (95.3)	85 (17.2)	
Household Income	<30,000 dollars	4554 (93.1)	334 (6.9)	0.000 **
	30,000 dollars=<	7474 (96.6)	242 (3.4)	
Sleeping Hours	<8 h per day	4987 (94.6)	287 (5.4)	0.724
	8 h per day=<	2819 (95.6)	152 (4.4)	
Obesity	Underweight	453 (94.2)	25 (5.8)	0.671
	Normal	5452 (95.3)	256 (4.7)	
	Obese	3151 (95.2)	152 (4.8)	
Subjective Health Condition	Good	3454 (98.2)	60 (1.8)	0.000 **
	Normal	6368 (96.5)	223 (3.5)	
	Bad	2234 (88.1)	294 (11.9)	
Stress Perception	No	8961 (96.9)	272 (3.1)	0.000 **
	Yes	3057 (90.8)	302 (9.2)	
Drinking Status	No	5483 (93.5)	383 (6.5)	0.000 **
	Yes	6001 (89.1)	734 (10.9)	
Current Smoking Status	No	9882 (95.2)	475 (4.8)	0.447
	Yes	2141 (95.7)	99 (4.3)	

** *p* < 0.001.

**Table 3 ijerph-18-00871-t003:** Socio-demographic variables and behavioral variables by self-reported depression symptom.

Characteristics	Sub-Items	Self-Reported Depression Symptom		
		No	Yes	*p*
Sex	Male	3460 (96.8)	133 (3.2)	0.000 **
	Female	4359 (93.5)	308 (6.5)	
Age	<30	955 (94.1)	63 (5.9)	
	30–39	1297 (95.4)	68 (4.6)	0.000 **
	40–49	1493 (96.5)	55 (3.5)	
	50–59	1472 (95.5)	70 (4.5)	
	60–69	1310 (94.1)	83 (5.9)	
	70=<	1292 (93.6)	102 (6.4)	
Marital Status	Living with a spouse	5504 (96.5)	218 (3.5)	0.000 **
	Living alone	1011 (89.3)	124 (10.7)	
Education	=<Middle school	2228 (92.1)	200 (7.9)	0.000 **
	High school=<	5588 (96.1)	240 (3.9)	
Region	Seoul	1587 (96.0)	66 (4.0)	0.025 *
	Metropolitan	2131 (94.3)	135 (5.7)	
	Gyeonggi-do province	1759 (95.9)	72 (4.1)	
	Other provinces except Gyeonggi-do	2342 (93.6)	168 (6.4)	
Experience of Basic Livelihood Protection	Yes	442 (86.1)	84 (13.9)	0.000 **
	No	7375 (95.5)	357 (4.5)	
Private Health Insurance	Yes	6170 (95.8)	271 (4.2)	0.000 **
	No	1605 (91.6)	170 (8.4)	
Type of Housing	Detached home	2555 (93.3)	195 (6.7)	0.000 **
	Apartment house	5264 (95.7)	246 (4.3)	
Health Insurance Coverage	Self-employed (insured)	2274 (93.5)	166 (6.5)	0.000 **
	Employee (insured)	5297 (96.2)	212 (3.8)	
	Medical Aid program	220 (82.3)	62 (17.7)	
Household Income	<30,000 dollars	2801 (91.8)	266 (8.2)	0.000 **
	30,000 dollars=<	5001 (96.7)	174 (3.3)	
Sleeping Hours	< 8 h per day	4987 (94.6)	287 (5.4)	0.117
	8 h per day=<	2819 (95.6)	152 (4.4)	
Obesity	Underweight	284 (91.5)	27 (8.5)	0.020 *
	Normal	3824 (95.3)	214 (4.7)	
	Obese	2161 (94.0)	130 (6.0)	
Subjective Health Condition	Good	2377 (99.1)	25 (0.9)	0.000 **
	Normal	4097 (96.7)	150 (3.3)	
	Bad	1344 (83.6)	266 (16.4)	
Stress Perception	No	5964 (98.4)	109 (1.6)	0.000 **
	Yes	1838 (85.0)	331 (15.0)	
Drinking Status	No	3623 (94.3)	229 (5.7)	0.040 *
	Yes	4079 (92.9)	311 (7.1)	
Current Smoking Status	No	6439 (95.4)	323 (4.6)	0.000 **
	Yes	1362 (92.6)	117 (7.4)	

* *p* < 0.05, ** *p* < 0.001.

**Table 4 ijerph-18-00871-t004:** Diagnosed depression and self-reported depression symptom.

Characteristics	Sub-Items	Diagnosed Depression		
		No	Yes	*p*
Self-Reported Depression Symptom	Yes	14 (13.1)	95 (86.9)	0.000 **
	No	137 (48.2)	154 (51.8)	

** *p* < 0.001.

**Table 5 ijerph-18-00871-t005:** Effects of socio-demographic variables, behavioral variables, and depression on the presence of periodontitis.

Characteristics		Model I		Model II	Model III		
		OR	95% CI	OR	95% CI	OR	95% CI
Sex	Male	1.690 **	1.463–1.954			1.587 **	1.296–1.820
	Female	1				1	
Age	<30	1				1	
	30–39	2.280 **	1.040–4.997			2.251	0.909–6.071
	40–49	4.500 **	2.097–9.658			4.739 **	1.856–12.010
	50–59	8.486 **	3.959–18.190			8.385 **	3.216–21.080
	60–69	9.196 **	4.257–19.866			9.228 **	3.974–23.289
	70 =<	7.917 **	3.596–17.432			8.441 **	3.690–22.221
Marital Status	Living with a spouse	0.734 *	0.606–0.889			0.709 *	0.573–0.884
	Living alone	1				1	
Education	=<Middle school	1.269 *	1.065–1.514			1.234 *	1.012–1.506
	High school=<	1				1	
Region	Seoul	0.700 *	0.546–0.897			0.657 **	0.499–0.872
	Metropolitan	0.882	0.706–1.102			0.951	0.723–1.232
	Gyeonggi-do province	0.600 **	0.468–0.768			0.594 **	0.453–0.777
	Other Provinces	1				1	
Experience of Basic Livelihood Protection	Yes	1.007	0.725–1.399			0.890	0.559–1.270
	No	1				1	
Private Health Insurance	Yes	0.845	0.703–1.017			0.819	0.672–1.064
	No	1				1	
Type of Housing	Detached home	1.235 *	1.029–1.482			1.278 *	1.038–1.621
	Apartment house	1				1	
Health Insurance coverage	Medical Aid program	1.487	0.935–2.364			0.987	0.554–1.716
	Employee (insured)	1.260	0.791–2.007			0.941	0.773–1.097
	Self-employed (insured)	1				1	
House Income	<30,000 dollars	1.159	0.991–1.355			1.150	0.972–1.427
	30,000 dollars=<	1				1	
Sleeping hours	<8 h per day	0.982	0.858–1.124			0.990	0.831–1.150
	8 h per day=<	1				1	
Obesity	Underweight	0.895	0.639–1.253			0.758	0.487–1.145
	Normal	1				1	
	Obese	1.262 *	1.104–1.443			1.311*	1.116–1.547
Subjective Health Condition	Good	1				1	
	Normal	1.004	0.850–1.185			0.964	0.757–1.130
	Bad	1.201	0.877–1.644			1.020	0.801–1.319
Stress Perception	No	1				1	
	Yes	1.186 *	1.007–1.397			0.951	0.801–1.134
Drinking Status	No	1				1	
	Yes	1.077	0.939–1.236			1.079	0.921–1.255
Current Smoking Status	No	1				1	
	Yes	2.143 **	1.783–2.575			2.210 **	1.772–2.763
Diagnosed Depression	No			1		1	
	Yes			1.443 **	1.099–1.898	1.772 **	1.328–2.364
Self–Reported Depression Symptom	No			1		1	
	Yes			1.133	0.857–1.497	1.250	0.875–1.787

* *p* < 0.05, ** *p* < 0.001.

## Data Availability

The authors have no authority over the data, and the data is provided upon request to the Ministry of Health and Welfare.

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
