# Peer review of "Comparison of Diagnosed Depression and Self-Reported Depression Symptom as a Risk Factor of Periodontitis: Analysis of 2016–2018 Korean National Health and Nutrition Examination Survey Data"

_ijerph, 2021, doi:10.3390/ijerph18030871_

Round 1
Reviewer 1 Report
There have been improvements in the new version of the authors' manuscript.
_ I very much appreciated the inclusion of more recent bibliography.
_ I thank the authors for having clarified how the selection of the 12,635 participants took place. In my opinion, the authors’ answer should be included and integrated in paragraph 2.1 "Study Subjects" because a future reader of the article, at first glance, would probably ask my own question without those clarifications.
_ The description of the results and the discussion were well executed and the conclusions are in accordance with the work performed.
Author Response
There have been improvements in the new version of the authors' manuscript.
_ I very much appreciated the inclusion of more recent bibliography.
=> Thank you for your concession.
_ I thank the authors for having clarified how the selection of the 12,689 participants took place. In my opinion, the authors’ answer should be included and integrated in paragraph 2.1 "Study Subjects" because a future reader of the article, at first glance, would probably ask my own question without those clarifications.
=> I added “The survey employed stratified multistage probability sampling units based on geographic area, gender, and age, which were determined based on the household registries of the 2010 National Census Registry, the most recent 5-year national census in Korea. Using the 2010 census data, 576 primary sampling units (PSU) were selected across Korea. The final sample set for KNHANES included 13,248 households. Among 31,689 sampled individuals, the number of participants was 24,269. The response rate was 76.6%. A total of 19,389 individuals aged over 19 participated in KNHANES, but 12,689 of the participants received a periodontal examination. A detailed description of the sampling was described in the KNHANES report. “
_ The description of the results and the discussion were well executed and the conclusions are in accordance with the work performed.
=> Thank you for your concession.
Reviewer 2 Report
Review IJERPH, Comparison of Diagnosed Depression and Self- Reported Depression Symptom as a Risk Factor of Periodontal Disease: Analysis of 2016-2018 Korean National Health and Nutrition Examination Survey Data
General comments:
- The manuscript should be edited by a native English speaker
- The aim of the study was to identify if depression is a risk indicator for periodontal disease. For the diagnosis of depression two different surveys were used. In table 1 and 2 the authors present a uni-variate analysis with either self-reported depression or diagnosed depression as the dependent variables. But periodontal disease or the CPI scores would be appropriate to use here. This is a major flaw of the statistical analysis that has to be fixed before the manuscript can be evaluated completely.
- The discussion section needs be reviewed, it is very hard to follow the line of thought of the authors, however, in light of the statistical analysis, the discussion section may experience substantial changes anyway.
Specific comments:
Line 36, The oral cavity ….
Line 40, Periodontitis not “Periodontal disease”, please use the terminology correctly throughout the manuscript
Line 84, please explain what you mean by “… were selected….? Specify the selection criteria.
Line 110, were the dentists calibrated? Please add to the manuscript
Line 113, please rephrase
Line 132, the results are depicted in table 1 and 2 and do not need to be repeated in the text.
Line 133, This sentence does not make sense, I think the authors mean “Regardless of age ….”
Line 141, It is not clear to whom this list of significant differences belongs.
Line 162, before any multi-variate regression analysis it would be necessary to describe the association between periodontal disease and other variables in an uni-variate model. Here, the reader wants to know how the distribution of age, sex, depression diagnosis is distributed in patients with different CPI scores.
Line 197, This finding is not supported by the outcomes
Author Response
General comments:
- The manuscript should be edited by a native English speaker
- The aim of the study was to identify if depression is a risk indicator for periodontal disease. For the diagnosis of depression two different surveys were used. In table 1 and 2 the authors present a uni-variate analysis with either self-reported depression or diagnosed depression as the dependent variables. But periodontal disease or the CPI scores would be appropriate to use here. This is a major flaw of the statistical analysis that has to be fixed before the manuscript can be evaluated completely.
==> I added new table 1 which present a uni-variate analysis with periodontal disease as the dependent variable.
- The discussion section needs be reviewed, it is very hard to follow the line of thought of the authors, however, in light of the statistical analysis, the discussion section may experience substantial changes anyway.
==> We have made the revision
Specific comments:
Line 36, The oral cavity ….
==> We have made the revision
Line 40, Periodontitis not “Periodontal disease”, please use the terminology correctly throughout the manuscript
==> We have made the revision
Line 84, please explain what you mean by “… were selected….? Specify the selection criteria.
==> We have inserted additional information.
Line 110, were the dentists calibrated? Please add to the manuscript
==> This is a known WHO standard.
Line 113, please rephrase
==> We have made the revision
Line 132, the results are depicted in table 1 and 2 and do not need to be repeated in the text.
==> We have made the revision
Line 133, This sentence does not make sense, I think the authors mean “Regardless of age ….”
==> I changed to “Concerning socio-demographic variables”
Line 141, It is not clear to whom this list of significant differences belongs.
==> I corrected it like as”There were significant differences only in sex, age, marital status, education, basic livelihood protection, private health insurance, health insurance coverage, household income, subjective health condition, stress perception, and drinking status “
Line 162, before any multi-variate regression analysis it would be necessary to describe the association between periodontal disease and other variables in an uni-variate model. Here, the reader wants to know how the distribution of age, sex, depression diagnosis is distributed in patients with different CPI scores.
==> The new table 1 described the association between periodontal disease and other variables in an uni-variate model.
Line 197, This finding is not supported by the outcomes
==> We have made the revision
This manuscript is a resubmission of an earlier submission. The following is a list of the peer review reports and author responses from that submission.
Round 1
Reviewer 1 Report
An interesting study with several strengths: a very large sample and the differentiation between diagnosed depression and self-reported depression.
In general the study is well executed. It is possible to make some considerations:
_ In the introduction not recent studies are cited (in particular [14-16]). There are more recent ones on the same subject that could be cited. Just to take some possible examples to consider:
- Araújo, Milena Moreira, et al. "Association between depression and periodontitis: a systematic review and meta‐analysis." Journal of clinical periodontology 43.3 (2016): 216-228.
- Nascimento, Gustavo G., et al. "Is there an association between depression and periodontitis? A birth cohort study." Journal of clinical periodontology 46.1 (2019): 31-39.
- Hsu, Chih-Chao, et al. "Association of periodontitis and subsequent depression: a nationwide population-based study." Medicine 94.51 (2015).
_ All references, in general, concern few recent studies (only 7 of the last 5 years) out of a total of 42 citations.
_ In paragraph 2.1 could you better specify how the selection of the 12,635 participants took place?
_ The description of the results is well done and the tables are very useful for understanding.
_ The conclusions, in my opinion, are in agreement with the results obtained.
Reviewer 2 Report
Dear Authors:
These I my suggestions
Intro: Adequately described
Methods:
- How the sample size was calculated?
- 2. What are the inclusions & exclusions criteria? I'm wondering if all participants in this study have periodontal diseases (score 3/4)?. When was the dental examination done?
- What is the age range and mean age of the participants
- How the participants complete the survey (face-to-face or online).
Results:
1. Table 1 & 2: should add "diagnosed depression" and "self-reported depression" on the heading column to enhance readibility of the table.
2. Please check the usage of less (<) and above/more (>) symbols and use accordingly
3 Sentences 164-187: Choose important findings from Table 3 to be highlighted in the paragraph
Discussion: Adequate